# A Systematic Review and Meta-Analysis of Mean Platelet Volume and Platelet Distribution Width in Patients with Obstructive Sleep Apnoea Syndrome

**DOI:** 10.3390/biomedicines12020270

**Published:** 2024-01-24

**Authors:** Biagio Di Lorenzo, Chiara Scala, Arduino A. Mangoni, Stefano Zoroddu, Panagiotis Paliogiannis, Pietro Pirina, Alessandro G. Fois, Ciriaco Carru, Angelo Zinellu

**Affiliations:** 1Department of Biomedical Sciences, University of Sassari, 07100 Sassari, Italycarru@uniss.it (C.C.); azinellu@uniss.it (A.Z.); 2Department of Medicine, Surgery and Pharmacy, University of Sassari, 07100 Sassari, Italyppaliogiannis@uniss.it (P.P.);; 3Clinical and Interventional Pulmonology, University Hospital of Sassari (AOU), 07100 Sassari, Italy; 4Discipline of Clinical Pharmacology, College of Medicine and Public Health, Flinders University, Bedford Park, SA 5042, Australia; 5Department of Clinical Pharmacology, Flinders Medical Centre, Southern Adelaide Local Health Network, Bedford Park, SA 5042, Australia; 6Quality Control Unit, University Hospital of Sassari (AOU), 07100 Sassari, Italy

**Keywords:** mean platelet volume, platelet distribution width, biomarker, obstructive sleep apnoea syndrome, disease severity, treatment

## Abstract

Obstructive sleep apnoea syndrome (OSAS) is a highly prevalent yet underestimated disorder caused by the complete or partial obstruction of the upper airways. Although polysomnography is the gold standard for OSAS diagnosis, there is an active search for easily accessible biomarkers of disease presence and severity, particularly those reflecting morphological changes in specific blood cells. We investigated the associations between the presence and severity of OSAS, continuous positive airway pressure (CPAP) treatment, mean platelet volume (MPV), and platelet distribution width (PDW), routinely assessed as part of the complete blood count. From 262 retrieved records from PubMed, the Web of Science, Scopus, and Google Scholar, 31 manuscripts were selected for a final analysis, 30 investigating MPV and 15 investigating PDW. MPV was not statistically different between OSAS patients and healthy controls; however, it progressively increased with disease severity. By contrast, OSAS patients had significantly higher PDW values than controls (SMD = 0.40, 95% CI: 0.25 to 0.56; *p* ˂ 0.001), and the difference increased with disease severity. In a univariate meta-regression, there were significant associations between the MPV and publication year, the apnoea–hypopnea index, and diabetes mellitus, while no associations were observed with the PDW. No significant between-group differences were observed in the subgroup analyses. These data suggest that PDW, and to a lesser extent, MPV, are potential biomarkers of OSAS and require further research to ascertain their pathophysiological significance (PROSPERO, CRD42023459413).

## 1. Introduction

Obstructive sleep apnoea syndrome (OSAS) is a highly prevalent yet underestimated disorder worldwide [1]: approximately 26% of adults in the U.S. [2] and half of studied cohorts from Chile and Russia [3,4] suffer from OSAS. Partial or complete obstruction of the upper airways during sleep, leading to episodes of apnoea or hypopnea, is the hallmark of the disease [5,6,7]. These events may result in metabolic changes, endothelial dysfunction, proinflammatory factor activation, and systemic oxidative stress, all of which raise the risk of cardiovascular disease [8,9,10,11,12]. In light of this, treatment is based on restoring physiological oxygen intake through continuous positive airway pressure (CPAP) therapy or surgical therapies addressing specific causative factors (i.e., maxillary expansion, maxillomandibular or jaw advancement surgery) [10,11]. The number of apnoeic or hypopneic occurrences per hour of sleep (apnoea–hypopnea index—AHI), is used to stratify the severity of OSAS into three categories—mild (5 ≤ AHI < 15), moderate (15 ≤ AHI < 30), and severe (AHI ≥ 30)—as determined in a polysomnography (PSG) sleep study [13], the standard diagnostic tool for OSAS. However, since PSG requires specialized diagnostic facilities, there is an open quest for the identification of alternative respiratory and/or circulating biomarkers to facilitate OSAS diagnosis and the assessment of treatment response [14,15,16,17,18,19,20]. Potential circulating biomarkers include platelet indices that are routinely assessed as part of the full blood count: the mean platelet volume (MPV, which reflects the platelet average volume and production) and the platelet distribution width (PDW, which reflects the size variation of platelets expressed as the percentage ratio between the platelet standard deviation and the MPV). The clinical relevance of the MPV and the PDW is supported by their critical role in coagulation, inflammation, thrombosis, and atherosclerosis [21]; the risk of cardiovascular disease (CVD), obesity, diabetes mellitus (DM), and hypertension (HPT) [22,23,24]; and other disease states, e.g., anaemia, thrombocytopenia, autoimmunity, cancer, preeclampsia, kidney, liver, and respiratory diseases [25,26,27,28,29,30,31,32,33,34,35,36,37,38,39,40,41,42,43,44,45,46,47,48]. However, given their uncertain role in OSAS, we carried out a systematic review and meta-analysis to look into potential correlations between the MPV and the PDW and the presence and severity of OSAS, as well as the outcomes of treatment.

## 2. Materials and Methods

### 2.1. Search Strategy, Eligibility Criteria, and Study Selection

We carried out a comprehensive search of publications using the following terms and their combinations in PubMed, the Web of Science, and Scopus from inception to 31 August 2023: “OSA”, “OSAS”, “obstructive sleep apnoea syndrome”, “complete blood count”, “CBC”, “full blood count”, “FBC”, “mean platelet volume”, “MPV”, “platelet distribution width”, and “PDW”. The full search strategy is reported in the Appendix A.

The full articles were independently reviewed by two researchers after establishing their relevance through abstract and title screening. The references of the retrieved articles were also hand-searched to identify additional studies. Any disagreement between the reviewers was settled by a third investigator. OSAS patients were categorized into mild, moderate, and severe OSAS (5 ≤ AHI < 15, 15 ≤ AHI < 30, and AHI ≥ 30, respectively). If available, MPVs and PDWs were also extracted following CPAP treatment. Subjects with simple snoring or AHI < 5 served as the control group.

The exclusion criteria were paediatric age (under 18), the concomitant presence of other diagnosed sleep disorders (e.g., central sleep apnoea), and corrective surgery.

The Joanna Briggs Institute (JBI) Critical Appraisal Checklist for analytical studies was used to assess the risk of bias. Studies were considered as having a low, moderate, or high risk of bias if they addressed more than 75%, between 50% and 75%, and less than 50% of the checklist items, respectively [49]. The Grades of Recommendation, Assessment, Development, and Evaluation (GRADE) Working Group guidelines were used to evaluate the certainty of evidence. The GRADE considers the study design (randomized vs. observational), the risk of bias (JBI checklist), the presence of unexplained heterogeneity, the indirectness of evidence, the imprecision of the results (sample size, 95% confidence interval width, and threshold crossing), the effect size (small, SMD < 0.5; moderate, SMD 0.5–0.8; and large, SMD > 0.8) [50], and the probability of publication bias [51,52]. The study complied with the Preferred Reporting Items for Systematic reviews and Meta-Analyses (PRISMA) 2020 statement [53]. The protocol was registered in the International Prospective Register of Systematic Reviews (PROSPERO, CRD42023459413).

### 2.2. Statistical Analysis

Standardized mean differences (SMDs) and 95% confidence intervals (CIs) were used to evaluate the differences in platelet indices between OSAS patients and control subjects and summarized in forest plots of continuous data. Values of *p* < 0.05 were considered statistically significant. Median and interquartile or min–max ranges were converted into mean and standard deviation as indicated by Wan et al. [54].

The heterogeneity of the SMDs across studies with a significance level set at *p* < 0.10 was assessed employing the Q statistic, while the inconsistency across studies was evaluated with the use of the I^2^ statistic (I^2^ < 25%, no heterogeneity; 25% ≤ I^2^ < 50%, moderate heterogeneity; 50% ≤ I^2^ < 75% large heterogeneity; and I^2^ > 75%, extreme heterogeneity) [55,56]. A random-effects model was used when I^2^ ≥ 50%.

To assess the impact of the MPV and the PDW extracted from each study on the overall risk estimate, we performed sensitivity analyses, where each study was sequentially removed [57]. The potential for publishing bias was estimated using the means of Begg’s adjusted rank correlation test and Egger’s regression asymmetry test at the *p* < 0.05 level of significance [58,59]. In the event of publication bias, the Duval and Tweedie “trim and fill” approach was carried out [60]. The effect sizes of the MPV and the PDW were examined for potential correlations with the year of publication, sample size, age, gender, AHI, body mass index (BMI), mean SpO2, min SpO2, study design, oxygen desaturation index (ODI), smoking status, DM, HPT, and CVD using univariate meta-regression and subgroup analyses. STATA 14 (STATA Corp., College Station, TX, USA) was utilized for conducting the statistical analysis.

## 3. Results

### 3.1. Systematic Research

From a total of 262 identified records through database searches, 133 were duplicates, while the remaining 129 were screened for title and abstract relevance. The full text of 69 eligible articles was further reviewed: 40 records were removed because of the lack of assessment of platelet indices (*n* = 11), the absence of a reference cohort (*n* = 7), the inclusion of paediatric patients (*n* = 6), the investigation of different patient cohorts (*n* = 7), the use of non-English language (*n* = 4), and non-clinical studies (*n* = 5), leaving 29 articles for data analysis. Two additional articles were identified by hand-searching these articles, yielding a total of 31 [8,19,61,62,63,64,65,66,67,68,69,70,71,72,73,74,75,76,77,78,79,80,81,82,83,84,85,86,87,88,89] (Figure 1). The risk of bias was low in all studies (Appendix A), while the cross-sectional design of the investigated studies led to a low initial level of the certainty of the evidence. (rating 2).

### 3.2. MPV

#### 3.2.1. Characteristics of the Included Studies

A total of thirty studies involving 6,560 OSAS subjects (mean age: 51 years, 72% male) and 1,724 healthy controls (mean age: 46 years, 61% male) were assessed [8,19,61,62,63,64,65,66,67,68,69,70,71,72,74,75,76,77,78,79,80,81,82,83,84,85,86,87,88,89] (Table 1). Twenty-two studies were conducted in Turkey [8,19,61,62,64,65,66,67,68,69,70,72,74,76,77,78,82,83,84,85,87,88], two in Egypt [71,81], two in China [80,89], two in Greece [63,79], one in South Korea [75], and one in Romania [86]. Nineteen studies had a retrospective design [8,62,65,66,67,71,72,74,75,76,77,78,80,84,85,87,88,89], whilst the remaining 11 were prospective [19,61,63,64,68,69,79,81,82,83,86].

#### 3.2.2. Results of Individual Studies and Syntheses

Appendix A shows the forest plot of the MPV values in OSAS patients and controls. Because of the significant heterogeneity observed, random-effects models were applied (I^2^ = 93.9%, *p* < 0.001). Overall, the pooled results indicated that there was no statistically significant difference in MPV values between OSAS patients and control subjects (SMD = 0.19, 95% CI: −0.04 to 0.43, *p* = 0.101). The sensitivity analysis revealed that when any individual study was removed, the corresponding pooled SMD values remained unchanged (effect size ranged between 0.12 and 0.26, Appendix A). No significant publication bias was detected using Begg’s (*p* = 0.27) or Egger’s test (*p* = 0.78). Accordingly, the “trim-and-fill” approach was unable to find any studies that needed to be added to the funnel plot in order to maintain symmetry (Appendix A).

#### 3.2.3. Meta-Regression and Analysis of Subgroups

Age (*t* = −1.10, *p* = 0.28), gender (*t* = −0.32, *p =* 0.75), BMI (*t* = −1.82, *p =* 0.081), mean SpO2 (*t* = 0.62, *p =* 0.55), min SpO2 (*t* = −1.59, *p =* 0.13), ODI (*t* = 0.14, *p =* 0.89), smoking (*t* = 0.11, *p =* 0.91), HPT (*t* = 0.10, *p =* 0.92), CVD (*t* = 1.72, *p =* 0.13), and sample size (*t* = −0.18, *p =* 0.86) were not significantly correlated with the MPV effect size. The SMD and publication year (*t* = −2.74, *p =* 0.01, Appendix A), AHI (*t* = 3.63, *p =* 0.002, Appendix A), and DM (*t* = −3.09, *p =* 0.01, Appendix A), on the other hand, all showed a significant correlation, also demonstrated through the cumulative analysis (Appendix A). The subgroup analysis did not show any significant difference (*p =* 0.29) between studies performed in Turkey (SMD = 0.11, 95% CI: −0.14 to 0.37, *p =* 0.39; I^2^ = 92.0%, *p* ˂ 0.001) or in other countries (SMD = 0.41, 95% CI: −0.09 to 0.90, *p =* 0.11; I^2^ = 96.3%, *p* ˂ 0.001, Appendix A). Similarly, no significant differences (*p =* 0.94) in pooled SMD were observed between retrospective (SMD = 0.19, 95% CI: −0.11 to 0.49, *p =* 0.22; I^2^ = 95.0%, *p* ˂ 0.001) and prospective studies (SMD = 0.21, 95% CI: −0.15 to 0.56, *p =* 0.26; I^2^ = 90.4%, *p* < 0.001, Appendix A).

#### 3.2.4. Certainty of Evidence

After taking into account the low risk of bias in all studies (no change), the high and unexplained heterogeneity (downgrade one level), the lack of indirectness (no change), the small effect size (SMD = 0.19, no change), and the absence of publication bias (no change), the overall level of certainty was downgraded to very low (rating one).

#### 3.2.5. MPV and Disease Severity

Eighteen studies [19,63,64,65,66,69,70,74,75,76,77,78,80,81,84,86,88,89] reported MPV values for mild disease, 19 for moderate disease [19,63,64,65,66,69,70,74,75,76,77,78,80,81,83,84,86,88,89], and 24 for severe disease [8,19,61,62,63,64,65,66,69,70,74,75,76,77,78,80,81,83,84,86,87,88,89]. The MPV values in OSAS subjects gradually increased compared to controls, from mild (SMD = 0.07; 95% CI: −0.11 to 0.24, *p =* 0.45; I^2^ = 75.8%, *p* < 0.001, Figure 2A) to moderate (SMD = 0.24; 95% CI: 0.00 to 0.49, *p =* 0.054; I^2^ = 88.3%, *p* < 0.001, Figure 2B) and to severe disease (SMD = 0.45; 95% CI: 0.18 to 0.72, *p =* 0.001; I^2^ = 93.2%, *p* < 0.001, Figure 2C). Additionally, severely ill OSAS patients had considerably higher MPV values than those with mild to moderate disease (SMD = 0.33, 95% CI: 0.13 to 0.52, *p =* 0.001; I^2^ = 91.3%, *p =* 0.001, Appendix A).

#### 3.2.6. MPV and CPAP Treatment

Five studies [8,62,82,83,86] also reported MPV values before and after CPAP treatment. Notably, a significant reduction was observed after treatment (SMD = −0.59, 95% CI: −1.02 to −0.16, *p =* 0.007; I^2^ = 78.5%, *p =* 0.001, Appendix A).

### 3.3. PDW

#### 3.3.1. Characteristics of the Included Studies

Fifteen studies with a total of 4899 OSAS patients (mean age: 50 years, 74% males) and 1132 healthy controls (mean age: 46 years, 60% males) from 15 studies were included in the evaluation of the PDW SMDs [8,63,66,68,69,70,71,72,73,74,79,80,83,88,89] (Table 2). Nine studies were conducted in Turkey [8,66,68,69,70,72,74,83,88], two in China [80,89], two in Greece [63,79], one in Egypt [71], and one in South Korea [73]. Ten studies had a retrospective design [61,69,70,71,72,73,74,80,88,89], whilst the remaining five were prospective [8,63,68,79,83].

#### 3.3.2. Results of Individual Studies and Syntheses

Figure 3 displays the forest plot for the PDW values in OSAS subjects and controls. The PDW levels were higher in all examined manuscripts describing OSAS patients in comparison with healthy subjects (mean difference range, 0.08 to 1.06). Because of the elevated heterogeneity amongst the studies (I^2^ = 78.7%, *p* ˂ 0.001), random-effects models were applied. PDW values were found to be substantially higher in OSAS participants than in controls according to the pooled data (SMD = 0.40, 95% CI: 0.25 to 0.56; *p* ˂ 0.001). The sensitivity analysis revealed that the corresponding pooled SMD values were not altered when any single study was in turn omitted (effect size ranged between 0.36 and 0.43, Appendix A). Neither Begg’s (*p* = 0.55) nor Egger’s tests (*p =* 0.54) revealed any evidence of publication bias. However, the “trim-and-fill” method identified one missing study to be added to the left side of funnel plot to ensure symmetry (Appendix A). The resulting effect size was similar to the primary analysis (SMD = 0.36, 95% CI: 0.19 to 0.52; *p* ˂ 0.001).

#### 3.3.3. Meta-Regression and Subgroup Analysis

No significant associations were observed in the univariate meta-regression analysis between the effect size and publication year (*t* = −2.11, *p =* 0.055), sample size (*t* = −1.57, *p =* 0.14), age (*t* = −0.12, *p =* 0.91), gender (*t* = −0.50, *p =* 0.63), AHI (*t* = 1.03, *p =* 0.33), BMI (*t* = −1.58, *p =* 0.14), mean SpO2 (*t* = −0.86, *p =* 0.42), min SpO2 (*t* = −1.08, *p =* 0.31), ODI (*t* = 0.46, *p =* 0.66), smoking (*t* = −1.90, *p =* 0.11), or HPT (*t* = 1.54, *p =* 0.17). In the subgroup analysis, non-significant differences (*p =* 0.47) were observed between studies conducted in Turkey (SMD = 0.35, 95% CI: 0.20 to 0.51, *p* ˂ 0.001; I^2^ = 47.4%, *p =* 0.055) and other countries (SMD = 0.48, 95% CI: 0.18 to 0.77, *p =* 0.001; I^2^ = 89.8%, *p* ˂ 0.001, Appendix A), with a significant decrease in the between-study variance in the first subgroup. In addition, no significant differences (*p =* 0.89) in the pooled SMDs were observed between retrospective (SMD = 0.41, 95% CI: 0.23 to 0.60, *p* ˂ 0.001; I^2^ = 80.4%, *p* ˂ 0.001) and prospective studies (SMD = 0.39, 95% CI: 0.11 to 0.67, *p =* 0.007; I^2^ = 78.7%, *p* < 0.001, Appendix A).

#### 3.3.4. Certainty of Evidence

The overall level of certainty remained low (rating two) after taking into consideration the low of bias in all studies (no change), the high but partially explainable heterogeneity (no change), the lack of indirectness (no change), the small effect size (SMD = 0.40, no change), and the absence of publication bias (no change).

#### 3.3.5. PDW and Disease Severity

Nine studies reported mild PDW values [63,66,69,70,73,74,80,88,89], ten were moderate [63,66,69,70,73,74,80,83,88,89], and 12 showed severe disease [8,63,66,69,70,71,73,74,80,83,88,89]. The PDW values increased gradually and were significantly higher in OSAS patients compared to controls, in a progressive manner, from mild (SMD = 0.29; 95% CI: 0.09 to 0.44, *p =* 0.004; I^2^ = 68.6%, *p =* 0.001, Figure 4A) to moderate (SMD = 0.36; 95% CI: 0.19 to 0.52, *p* ˂ 0.001; I^2^ = 60.9%, *p =* 0.006, Figure 4B) and to severe disease (SMD = 0.50; 95% CI: 0.25 to 0.75, *p* ˂ 0.001; I^2^ = 88.2%, *p* < 0.001, Figure 4C). The PDW values were also significantly higher in subjects with the severe form of the disease than patients with mild/moderate OSAS (SMD = 0.24, 95% CI: 0.07 to 0.41, *p =* 0.001; I^2^ = 84.1%, *p =* 0.006, Appendix A).

#### 3.3.6. PDW and CPAP Treatment

Only two studies [8,83] reported RDW values before and after CPAP treatment with non-significant differences (SMD = 0.32, 95% CI: −0.04 to 0.68, *p =* 0.08; I^2^ = 45.5%, *p =* 0.175, Appendix A).

## 4. Discussion

Several candidates have been proposed as OSAS biomarkers [17,20]; however, the search for the ideal OSAS circulating biomarker is still at an early stage. In our systematic review and metanalysis, we aimed to investigate possible alterations in platelet indices, MPV and PDW, in OSAS, as well as to determine additional associations with disease severity and CPAP treatment.

Although we observed no significant differences in MPV between OSAS patients and controls, there were significant associations between the MPV SMDs and the year of publication, DM, and AHI. The correlation of the AHI with the MPV SMDs could be explained by the fact that the MPVs increased with increasing disease severity. However, the observation that the MPV SMDs decreased with the year of publication during the period of 2010–2023 is surprising, as there were no significant temporal differences in the laboratory methods that could account for this observation. Similarly, the observed reduction in the MPV SMDs that correlated with the presence of DM is counterintuitive and requires further study.

Nevertheless, the stratification based on disease severity highlighted that MPV values progressively increase from mild to moderate and severe forms. Additionally, given the well-known positive effect of CPAP on oxygen saturation and apnoea in patients affected by respiratory disorders [90,91], we collected the available information on the impact of CPAP on MPV. Four of five selected manuscripts [8,62,82,86] described a positive effect of CPAP treatment: already, after two months of treatment, the MPV values of OSAS subjects were similar to non-OSAS individuals [86].

Regarding PDW, we observed that the pooled SMD was significantly higher in OSAS patients when compared to controls. Additionally, the PDW values increased with disease severity. However, only two studies reported the effect of CPAP on PDW [8,83], and therefore, additional research is warranted to investigate this issue.

MPV and PDW are markers of platelet activation, and their increase could theoretically reflect the clinical manifestations of OSAS. However, the mechanisms underlying platelet activation in OSAS are still matter of active research, with accumulating evidence suggesting a complex and synergistic network of molecular mechanisms. For instance, platelets could be activated in a dose-dependent manner by circulating catecholamines in response to sympathetic activation as well as chronic intermittent hypoxia, two typical traits of OSAS subjects [92,93,94]. In turn, higher levels of platelet activation could contribute to the increased risk of atherosclerosis, cardiovascular disease, and other comorbidities in this patient group [92,95].

Despite the promising role of MPV and PDW as biomarkers of the presence and severity of OSAS, a high to extreme between-study heterogeneity was observed for both parameters. Nevertheless, no evidence of publication bias was observed for MPV, while a missing study to ensure symmetry was required in the PDW funnel plot. However, adding a hypothetical study did not alter the PDW SMD, highlighting the consistency of the primary analysis.

An additional strength of our work is represented by the assessment of the GRADE certainty of evidence and by the conduct of meta-regression and subgroup analyses, which did not identify significant differences in effect size according to the study location or design, although limited information was available in European, American, and African patient populations. In addition to the high heterogeneity observed, which could be partially explained by the correlation with the publication year in meta-regression analysis (MPV) and the limited number of studies conducted outside the Middle East and Asia, the main limitation is represented by the small number of longitudinal studies and those investigating the effect of CPAP.

## 5. Conclusions

Our study showed that while MPV values in OSAS patients are comparable to non-OSAS subjects, PDW is significantly increased in OSAS. However, both platelet indices are positively related to disease severity. These results suggest the existence of an underlying pathogenic process that is most likely connected to alterations in inflammation and oxygenation, together with the decline in the mean SpO2. However, future studies seeking to confirm the relationship between MPV, PDW, and OSAS should be carried out, utilizing standardized procedures and diagnostic criteria, given the high heterogeneity in the included studies.

## Figures and Tables

**Figure 1 biomedicines-12-00270-f001:**
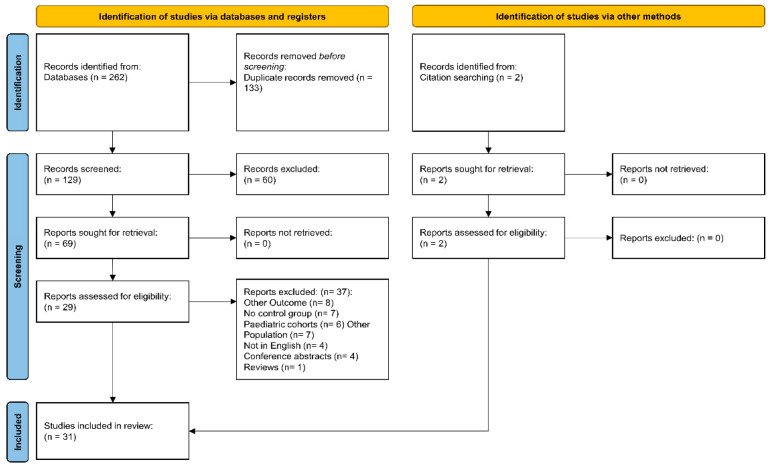
PRISMA 2020 flow diagram for systematic reviews.

**Figure 2 biomedicines-12-00270-f002:**
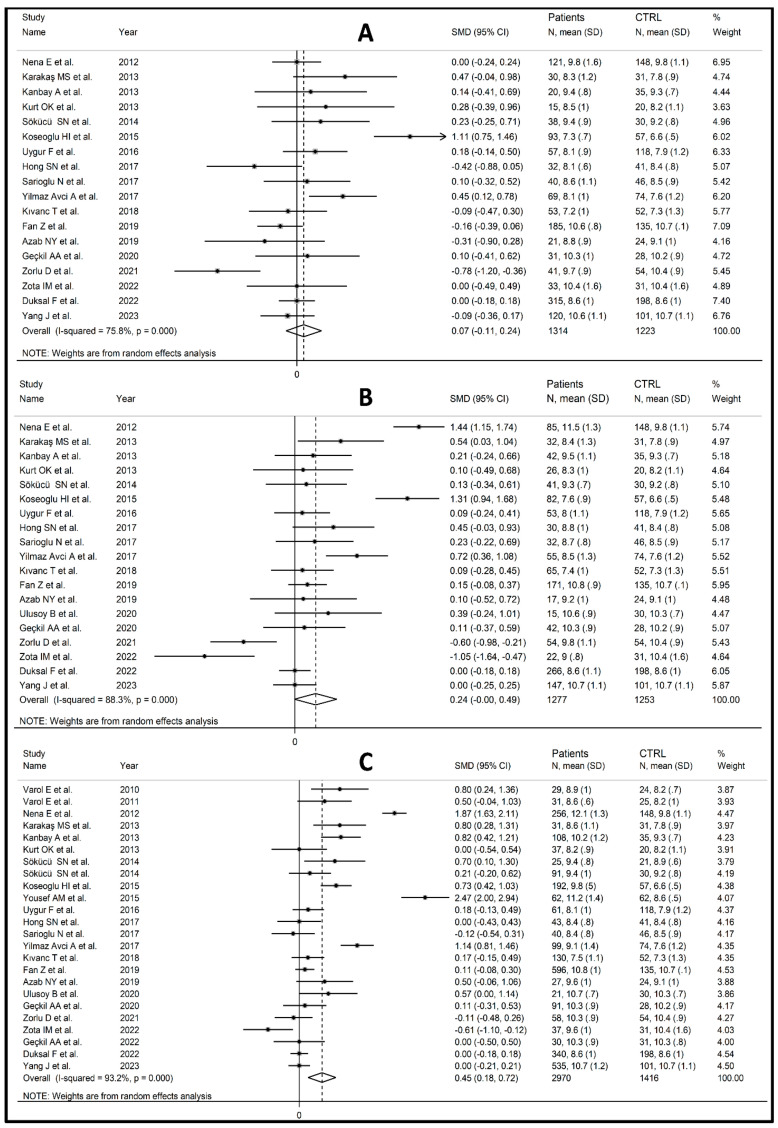
Forest plot of studies assessing MPV values of mild [19,63,64,65,66,69,70,74,75,76,77,78,80,81,84,86,88,89] (**A**), moderate [19,63,64,65,66,69,70,74,75,76,77,78,80,81,83,84,86,88,89] (**B**), and severe OSAS patients [8,19,61,62,63,64,65,66,69,70,74,75,76,77,78,80,81,83,84,86,87,88,89] (**C**) compared to controls.

**Figure 3 biomedicines-12-00270-f003:**
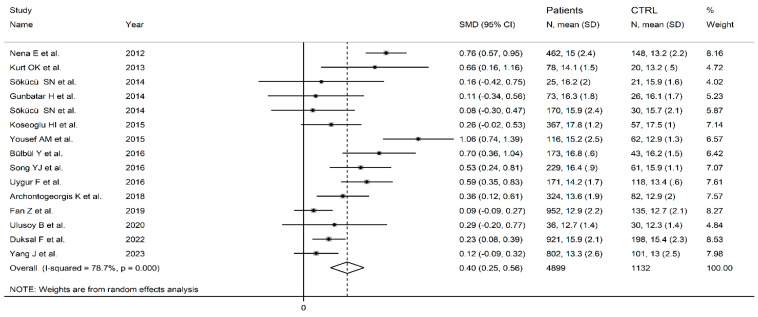
Forest plot of studies examining PDW values of OSAS patients and controls [8,63,66,68,69,70,71,72,73,74,79,80,83,88,89].

**Figure 4 biomedicines-12-00270-f004:**
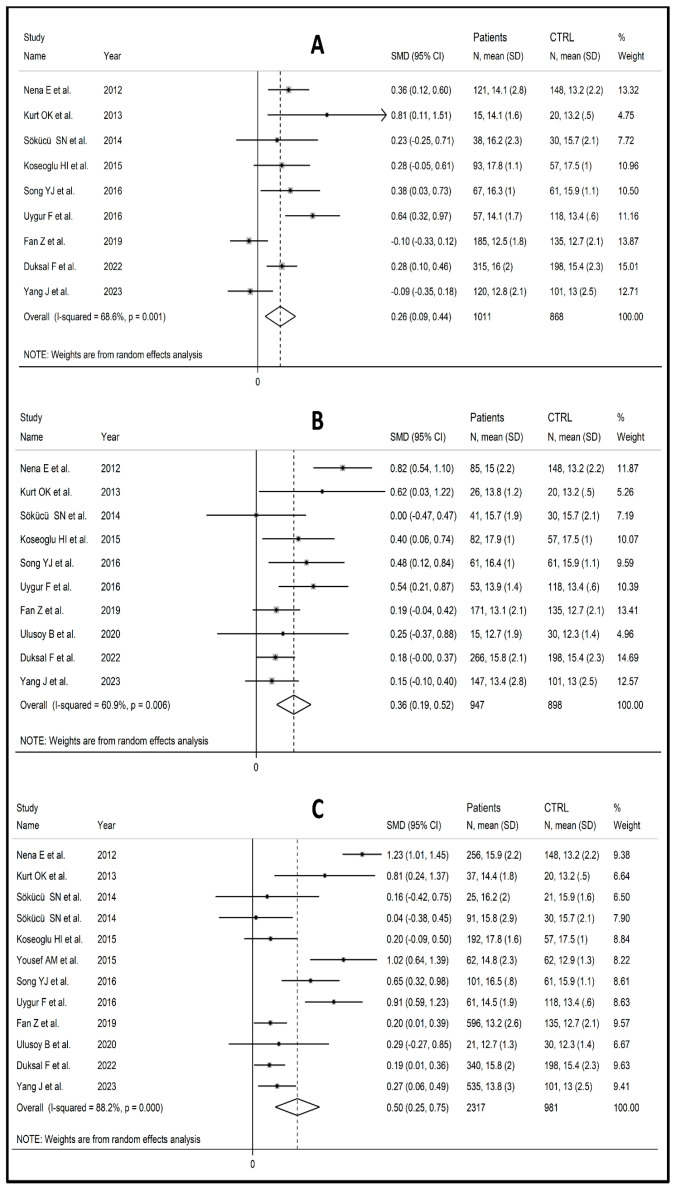
Forest plot of studies examining PDW values of mild [63,66,69,70,73,74,80,88,89] (**A**), moderate [63,66,69,70,73,74,80,83,88,89] (**B**), and severe OSAS patients [8,63,66,69,70,71,73,74,80,83,88,89] (**C**) compared to controls.

**Table 1 biomedicines-12-00270-t001:** Summary of studies collecting evidence on MPV (n = 30) with related demographics and study-related variables. Values are indicated as mean ± standard deviation (SD), and study design is expressed as P (prospective) or R (retrospective).

Study Name	Year	Country	Study Design	Non-OSAS				OSAS			
				Sample Size	Age	Gender (M/F)	MPV	Sample Size	Age	Gender (M/F)	MPV
					(Mean ± SD)		(Mean ± SD)		(Mean ± SD)		(Mean ± SD)
Varol, E. [61]	2010	Turkey	P	24	45.6 ± 13.9	14/10	8.2 ± 0.7	71	49.9 ± 9.8	43/28	8.7 ± 0.9
2. Varol, E. [62]	2011	Turkey	R	25	49.6 ± 8.5	14/11	8.2 ± 1	31	53.8 ± 9.2	21/10	8.6 ± 0.6
3. Nena, E. [63]	2012	Greece	P	148	-	-	9.8 ± 1.1	462	-	-	11.1 ± 1.4
4. Karakaş, M.S. [64]	2013	Turkey	P	31	46.7 ± 8.4	-	7.8 ± 0.9	93	47.2 ± 7.8	-	8.4 ± 1.2
5. Kanbay, A. [65]	2013	Turkey	R	35	51.2 ± 12.6	22/13	9.3 ± 0.7	170	54.7 ± 12.3	81/89	9.7 ± 1
6. Kurt, O.K. [66]	2013	Turkey	R	20	46.3 ± 13.1	11/9	8.2 ± 1.1	78	54.6 ± 10.7	51/27	8.3 ± 1
7. Sökücü, S.N. [8]	2014	Turkey	P	21	40.8 ± 11.6	21/0	8.9 ± 0.6	25	47.4 ± 11.7	25/0	9.4 ± 0.8
8. Akyüz, A. [67]	2014	Turkey	R	30	44 ± 11	21/9	8.8 ± 0.8	52	48 ± 11	31/21	8.5 ± 1.1
9. Gunbatar, H. [68]	2014	Turkey	P	26	41.3 ± 11	-	9.1 ± 1.5	73	50.8 ± 11.7	-	8.6 ± 0.9
10. Sökücü, S.N. [69]	2014	Turkey	R	30	38.4 ± 12.8	15/15	9.2 ± 0.8	170	45.5 ± 11.1	139/31	9.4 ± 0.9
11. Koseoglu, H.I. [70]	2015	Turkey	R	57	43.5 ± 11.2	23/34	6.6 ± 0.5	367	51.3 ± 10	259/108	8.2 ± 2.2
12. Yousef, A.M. [71]	2015	Egypt	R	62	52.3 ± 4.1	-	8.6 ± 0.5	116	52.8 ± 6.1	-	11.2 ± 1.3
13. Bülbül, Y. [72]	2016	Turkey	R	43	42.3 ± 10.5	18/25	8.5 ± 1.2	173	53.9 ± 10.8	105/68	8.4 ± 1
14. Uygur, F. [74]	2016	Turkey	R	118	50.3 ± 11.7	61/57	7.9 ± 1.2	171	53.3 ± 11.9	105/66	8.1 ± 1
15. Hong, S.N. [75]	2017	South Korea	R	41	28.9 ± 10.5	31/10	8.4 ± 0.8	105	37.1 ± 10.6	104/1	8.4 ± 0.8
16. Sarioglu, N. [76]	2017	Turkey	R	46	41.7 ± 9	31/15	8.5 ± 0.9	112	45.8 ± 9.5	86/26	8.6 ± 0.9
17. Yilmaz Avci, A. [77]	2017	Turkey	R	74	44 ± 13	37/37	7.6 ± 1.2	223	55 ± 12.7	158/65	8.6 ± 1.2
18. Kıvanc, T. [78]	2018	Turkey	R	52	41 ± 12	35/17	7.3 ± 1.3	248	48 ± 10.7	192/56	7.4 ± 1
19. Archontogeorgis, K. [79]	2018	Greece	P	82	47.2 ± 13.3	60/22	10 ± 0.9	324	53.7 ± 12.5	234/90	10.3 ± 1.2
20. Fan, Z. [80]	2019	China	R	135	46.3 ± 12	-	10.7 ± 0.1	952	44.8 ± 11	-	10.7 ± 0.9
21. Azab, N.Y. [81]	2019	Egypt	P	24	47.9 ± 11.5	15/9	9.1 ± 1	65	49.4 ± 11.7	36/29	9.3 ± 1
22. Lakadamyali, H. [82]	2019	Turkey	P	24	47.9 ± 13.2	16/8	7.7 ± 1.5	46	52.2 ± 12.7	38/8	8 ± 1.3
23. Ulusoy, B. [83]	2020	Turkey	P	30	42.3 ± 8.6	16/14	10.3 ± 0.7	36	42.5 ± 9.5	29/7	10.7 ± 1.4
24. Geçkil, A.A. [84]	2020	Turkey	R	28	54 ± 13.2	16/12	10.2 ± 0.9	164	55.3 ± 9.4	79/85	10.3 ± 0.9
25. Zorlu, D. [19]	2021	Turkey	P	54	53.1 ± 11.2	26/28	10.4 ± 0.9	153	54.3 ± 11.9	95/58	9.9 ± 1
26. Şeyhanlı, E.S. [85]	2021	Turkey	R	103	46.9	-	10.1 ± 1.1	208	54.4	-	8 ± 1.3
27. Zota, I.M. [86]	2022	Romania	P	31	49.55 ± 14	16/15	10.4 ± 1.6	92	56.8 ± 11.4	62/30	9.7 ± 1.1
28. Geçkil, A.A. [87]	2022	Turkey	R	31	41.1 ± 9.4	-	10.3 ± 0.8	57	50.7 ± 13.3	-	10.3 ± 0.9
29. Duksal, F. [88]	2022	Turkey	R	198	42.28 ± 12.1	101/97	8.6 ± 1	921	50 ± 11.4	534/387	8.6 ± 1.1
30. Yang, J. [89]	2023	China	R	101	51.3 ± 9.8	101/0	10.7 ± 1.1	802	54.4 ± 9.7	802/0	10.7 ± 1.1
Total				1724	45.44 ± 11.19	721/467	8.98 ± 0.95	6560	50.61 ± 10.79	3309/1290	9.20 ± 1.08

**Table 2 biomedicines-12-00270-t002:** Summary of studies collecting evidence on PDW (n = 15) with related demographics and study-related variables. Values are indicated as mean ± standard deviation (SD), and study design is expressed as P (prospective) or R (retrospective).

Study Name	Year	Country	Study Design	Non-OSAS				OSAS			
				Sample Size	Age	Gender (M/F)	PDW	Sample Size	Age	Gender (M/F)	PDW
					(Mean ± SD)		(Mean ± SD)		(Mean ± SD)		(Mean ± SD)
1. Nena, E. [63]	2012	Greece	P	148	-	-	13.2 ± 2.2	462	-	-	15 ± 2.4
2. Kurt, O.K. [66]	2013	Turkey	R	20	46.3 ± 13.1	09-Nov	13.2 ± 0.5	78	54.6 ± 10.7	51/27	14.1 ± 1.5
3. Sökücü, S.N. [8]	2014	Turkey	P	21	40.8 ± 11.6	21/0	15.9 ± 1.6	25	47.4 ± 11.7	25/0	16.2 ± 2
4. Gunbatar, H. [68]	2014	Turkey	P	26	41.3 ± 11	-	16.1 ± 1.7	73	50.8 ± 11.7	-	16.3 ± 1.8
5. Sökücü, S.N. [69]	2014	Turkey	R	30	38.4 ± 12.8	15/15	15.7 ± 2.1	170	45.5 ± 11.1	139/31	15.9 ± 2.4
6. Koseoglu, H.I. [70]	2015	Turkey	R	57	43.5 ± 11.2	23/34	17.5 ± 1	367	51.3 ± 10	259/108	17.8 ± 1.2
7. Yousef, A.M. [71]	2015	Egypt	R	62	52.3 ± 4.1	-	12.9 ± 1.3	116	52.8 ± 6.1	-	15.2 ± 2.5
8. Bülbül, Y. [72]	2016	Turkey	R	43	42.3 ± 10.5	18/25	16.2 ± 1.5	173	53.9 ± 10.8	105/68	16.8 ± 0.6
9. Song, Y.J. [73]	2016	South Korea	R	61	44 ± 15.4	33/28	15.9 ± 1.1	229	49.6 ± 12.6	160/69	16.4 ± 0.9
10. Uygur, F. [74]	2016	Turkey	R	118	50.3 ± 11.7	61/57	13.4 ± 0.6	171	53.3 ± 11.9	105/66	14.2 ± 1.7
11. Archontogeorgis, K. [79]	2018	Greece	P	82	47.2 ± 13.3	60/22	12.9 ± 2	324	53.7 ± 12.5	234/90	13.6 ± 1.9
12. Fan, Z. [80]	2019	China	R	135	46.3 ± 12	-	12.7 ± 2.1	952	44.8 ± 11	-	12.9 ± 2.2
13. Ulusoy, B. [83]	2020	Turkey	P	30	42.3 ± 8.6	16/14	12.3 ± 1.4	36	42.5 ± 9.5	29-Jul	12.7 ± 1.4
14. Duksal, F. [88]	2022	Turkey	R	198	42.28 ± 12.1	101/97	15.4 ± 2.3	921	50 ± 11.4	534/387	15.9 ± 2.1
15. Yang, J. [89]	2023	China	R	101	51.3 ± 9.8	101/0	13 ± 2.5	802	54.4 ± 9.7	802/0	13.3 ± 2.6
Total				1132	44.90 ± 11.23	460/301	14.42 ± 1.59	4899	50.33 ± 10.76	2433/853	15.09 ± 1.81

## Data Availability

All data relevant to the study are included in the article.

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
