# Peer review of "A Systematic Review and Meta-Analysis of Mean Platelet Volume and Platelet Distribution Width in Patients with Obstructive Sleep Apnoea Syndrome"

_biomedicines, 2024, doi:10.3390/biomedicines12020270_

Round 1

Reviewer 1 Report

Comments and Suggestions for Authors

The discussion should commence not with general conclusions, but with the conclusions drawn by the authors.

In some of the selected studies, the number of cases is quite low, which presents challenges in interpreting the results.

Author Response

Re: biomedicines-2789501

Dear Editor and Reviewers, 

Many thanks for your useful feedback regarding our manuscript. We wish to submit a revised version that takes into account the comments by the Reviewers and the Editor. All Authors have actively contributed to preparing the response and the revised version and agree with its contents. We declare no conflict of interest. Our responses to the points raised are as follow:

REVIEWER 1

We would like to thank the reviewer for her/his time and positive feedback.

The discussion should commence not with general conclusions, but with the conclusions drawn by the authors

As requested, we have rephrased the discussion which now commences with the study conclusions. 

In some of the selected studies, the number of cases is quite low, which presents challenges in interpreting the results.

We agree with the Reviewer that the number of cases in some studies is relatively low, an aspect which might present challenges in the interpretation of the results. However, there was no publication bias with the Begg’s and the Egger’s test, which evaluate the impact of studies with small populations. Additionally, no case reports nor case series were included in the list of selected studies, eliminating a source of potential bias and increasing the robustness of our analyses. Finally, the sensitivity and the “trim and fill” analyses performed to rule out the impact of each individual study on the SMD and to assess the “presence of missing studies”, did not reveal any significant impact of the included studies on our data, with the exception of one missing study to be added to the left side of PDW funnel plot to ensure symmetry (Suppl. Figure 12). We think that these considerations support the strength of our conclusions.

Best regards,

Biagio Di Lorenzo, on behalf of all contributing Authors. 

Reviewer 2 Report

Comments and Suggestions for Authors

This manuscript is a Systematic Review and Meta-Analysis of Mean Platelet Volume and Platelet Distribution Width in Patients with Obstructive Sleep Apnoea Syndrome.

It is an interesting and well-written paper.

I recommend its publication after some minor improvements:

-       Explain why so many duplicates were found: 133 out of 262 works?

-       Explain why so many Turkish papers were included: 22?

-       Discussion: Avoid starting discussing other biomarkers not at all addressed in this manuscript (lines 253-8).

-       Add a paragraph on limitations of the study at the end of discussion.

-       In conclusions, remove any physiopathological discussion not addressed in this work.

Author Response

Re: biomedicines-2789501

Dear Editor and Reviewers, 

Many thanks for your useful feedback regarding our manuscript. We wish to submit a revised version that takes into account the comments by the Reviewers and the Editor. All Authors have actively contributed to preparing the response and the revised version and agree with its contents. We declare no conflict of interest. Our responses to the points raised are as follow:

REVIEWER 2

This manuscript is a Systematic Review and Meta-Analysis of Mean Platelet Volume and Platelet Distribution Width in Patients with Obstructive Sleep Apnoea Syndrome.

It is an interesting and well-written paper.

We would like to thank the reviewer for her/his positive feedback and for the work, comments and precious suggestions. We will reply point-by-point hereby. 

I recommend its publication after some minor improvements:

-       Explain why so many duplicates were found: 133 out of 262 works?

We found 133 duplicated records on a total of 262 (about 50%) because the search was performed in three different databases (Pubmed, Scopus, and Web of Sciences). Please refer to paragraph 2.1 for the study selection and inclusion process. Therefore, the retrieved records were further screened for inter-database duplicates to see how many of them were found to be overlapping across different databases.

-       Explain why so many Turkish papers were included: 22?

We included all the manuscripts that were eligible, please refer to the methods section or to the review protocol on PROSPERO – CRD42023459413. We also noticed that around 70% of the included studies were conducted in Turkey, which might be due to at least two reasons: 1) a higher prevalence of OSAS in the Middle East region and 2) a tendency to conduct low-budget but high-impact studies using from routinely avaiable laboratory tests. To overcome this “geographic bias”, we ran a subgroup analysis stratifying the studies performed in Turkey vs. other countries, to see if this bias could somehow affect our results. This analysis did not reveal any significant between-group difference in either the MPV SMD or the PDW SMD (Suppl. Figure 7 and suppl. Figure 13, respectively). We and other research groups have addressed this issue in previous studies (doi: 10.3390/jcm12093302 and doi: 10.21037/jtd.2018.10.105) where publications from Turkey accounted for 75% and 82% of the selected studies, respectively.. 

-       Discussion: Avoid starting discussing other biomarkers not at all addressed in this manuscript (lines 253-8).

The initial discussion paragraph has been rephrased to “Several candidates have been proposed as OSAS biomarkers [17,20], however, the search for the ideal OSAS circulating biomarker is still at an early stage.” (lines 253ff).

-       Add a paragraph on limitations of the study at the end of discussion.

The limitations described in discussion are now highlighted (lines 300ff).

-       In conclusions, remove any physiopathological discussion not addressed in this work.

The links to other pathologies in the discussion have now been removed (lines 280ff). 

Additionally, please find enclosed a revised version of the manuscript.

Best regards,

Biagio Di Lorenzo, on behalf of all contributing Authors.